# Pancreatic Islet Viability Assessment Using Hyperspectral Imaging of Autofluorescence

**DOI:** 10.3390/cells12182302

**Published:** 2023-09-19

**Authors:** Jared M. Campbell, Stacey N. Walters, Abbas Habibalahi, Saabah B. Mahbub, Ayad G. Anwer, Shannon Handley, Shane T. Grey, Ewa M. Goldys

**Affiliations:** 1Australian Research Council Centre of Excellence for Nanoscale BioPhotonics, Graduate School of Biomedical Engineering, University of New South Wales, Sydney, NSW 2033, Australia; a.habibalahi@unsw.edu.au (A.H.); saabah.mahbub@gmail.com (S.B.M.); a.anwer@unsw.edu.au (A.G.A.); shannon.handley@student.unsw.edu.au (S.H.); e.goldys@unsw.edu.au (E.M.G.); 2Garvan Institute of Medical Research, Faculty of Medicine, St Vincent’s Clinical School, University of New South Wales, Sydney, NSW 2052, Australia; s.walters@garvan.org.au (S.N.W.); s.grey@garvan.org.au (S.T.G.)

**Keywords:** islet, hyperspectral, multispectral, autofluorescence, viability, transplantation

## Abstract

Islets prepared for transplantation into type 1 diabetes patients are exposed to compromising intrinsic and extrinsic factors that contribute to early graft failure, necessitating repeated islet infusions for clinical insulin independence. A lack of reliable pre-transplant measures to determine islet viability severely limits the success of islet transplantation and will limit future beta cell replacement strategies. We applied hyperspectral fluorescent microscopy to determine whether we could non-invasively detect islet damage induced by oxidative stress, hypoxia, cytokine injury, and warm ischaemia, and so predict transplant outcomes in a mouse model. In assessing islet spectral signals for NAD(P)H, flavins, collagen-I, and cytochrome-C in intact islets, we distinguished islets compromised by oxidative stress (ROS) (AUC = 1.00), hypoxia (AUC = 0.69), cytokine exposure (AUC = 0.94), and warm ischaemia (AUC = 0.94) compared to islets harvested from pristine anaesthetised heart-beating mouse donors. Significantly, with unsupervised assessment we defined an autofluorescent score for ischaemic islets that accurately predicted the restoration of glucose control in diabetic recipients following transplantation. Similar results were obtained for islet single cell suspensions, suggesting translational utility in the context of emerging beta cell replacement strategies. These data show that the pre-transplant hyperspectral imaging of islet autofluorescence has promise for predicting islet viability and transplant success.

## 1. Introduction

The transplantation of pancreatic islets is effective for patients with T1D and hyperglycaemic unawareness [1], however its efficiency is low, with most patients requiring high islet numbers, over multiple islet infusions (transplants), to become insulin-independent [2,3]. Although transplantation failure is multifactorial, it has been linked to islet viability [3], highlighting the need for methods to characterise islet quality prior to transplantation to inform clinical decisions [4]. However, common methodologies for assessing islet viability, including the consideration of islet morphology by a manual observer, live/dead assays (chiefly via membrane impermeable fluorescent dyes), ATP/ADP ratio, and glucose-stimulated insulin secretion (GSIS), are not strongly predictive of insulin independence after transplant [4,5,6,7,8,9]. Hypothesised causes of this include the measurement of living versus dead cells pre-transplantation not capturing apoptotic and pre-apoptotic cells, as well as quiescent β-cells recovering activity once reintroduced to physiological conditions [4]. Oxygen consumption showed promising accuracy but has not achieved clinical implementation [10]. 

Inflammation and oxidative damage (reactive oxygen species (ROS)-induced) triggered through islet isolation can impact β-cell metabolism, potentially affecting islet viability and, later, graft function [11,12,13]. The reduced form of nicotinamide adenine dinucleotide (NADH) and oxidised flavin adenine dinucleotide (FAD), the principal electron donors and acceptors of oxidative phosphorylation [14], are key indicators of metabolic state. The relative concentrations of these coenzymes—determined by a measure termed the optical redox ratio—has been linked to apoptosis [15], neoplasia [16], and stem cell differentiation [17]. NADH and FAD are autofluorophores, with NADH having excitation maxima at 290 and 351 nm and emission maxima at 440 and 460 nm, and FAD having an excitation maxima at 450 nm and its emission maxima at 535 nm [18]. As such, the relative abundance of these factors can be detected by their defined spectral signatures, allowing a real-time, sensitive readout of the cellular redox state, which, in turn, reflects cell viability and function [15,16,17].

Here, we have applied hyperspectral imaging technology to determine islet autofluorescence following isolation under optimal (heart beating anesthetised mouse donor) conditions. A broad light spectrum was analysed to capture data from an extensive cross-section of autofluorescent molecules within the cell—not just NAD(P)H and flavins, which are the typical targets of autofluorescent microscopy—giving a deep signature of biological status. Systems that have been characterised by this methodology include cell cycle stage [19], arthritic cartilage [20], kidney function [21], levels of ROS [22], and age [23]. The spectral signature of pristine pancreatic islets was then compared to islets exposed to viability-compromising insults—warm ischaemia, hypoxia, oxidative stress, and cytokine injury—in order to investigate whether a spectral signature could be defined to detect exposure to these sources of damage. As a secondary objective, we unmixed the spectra in order to study changes in specific component autofluorophores. Furthermore, we investigated whether a signature could be developed to predict post-transplant graft function in diabetic recipients (mouse). A non-invasive, label-free assay to assess the functional viability of islet preparations would provide an evidence-based platform for determining specific islet preparations’ likelihood of post-transplant success. This would improve patient experiences by avoiding the burden of unsuccessful procedures and could contribute to increasing the rate of insulin independence in T1D patients whilst reducing the need for multiple islet infusions. 

## 2. Materials and Methods

### 2.1. Islet Collection and Culture

Pancreatic islets from 2 to 4 C57BL/6Ausb mice (Australian BioResources (Moss Vale, NSW, Australia)) per experiment (mixed sex) were harvested according to our protocol [24], exposed to a compromising intervention, then imaged on a hyperspectral microscope. Ethics approval was given by the Garvan Institute of Medical Research Animal Research Authority (20_18). All culture was carried out at 37 °C and 5% CO_2_ in islet culture media [24]. Interventions included ROS clearance inhibitor menadione (30 µM) [25], hypoxia inductor dimethyloxalylglycine (DMOG) [11], pro-inflammatory cytokines (200 U/µL TNF-α, 200 U/µL IFNγ, and 25 U/µL IL-1β [26,27]), or warm ischaemia (30 and 60 min delayed pancreas collection [28]). Menadione treatment was 2 h of exposure (in culture media) followed by 24 h culture. DMOG treatment was 0.5 mM/L for 16 h prior to immediate imaging. Cytokine exposure was 4 or 24 h in islet culture media to induce moderate or major inflammation, respectively. Control islets were maintained in culture media with equivalent solvent used to emulsify interventions as applicable for identical time-courses. Similarly, islets for the warm ischaemia experiment were maintained in culture for 24 h before imaging. For the ROS, hypoxia, and inflammatory models, islets from different mice were pooled prior to being sorted into the treatment groups, ensuring that differences were not the consequence of animal or isolation factors. This was not possible for warm ischaemia, as the intervention was applied at the animal level. Three mice were used for each treatment group to accommodate this. Disaggregation of islets to individual cells was performed using 0.5 mM EDTA. 

Islet encapsulation was investigated to determine whether the hyperspectral technology could still be applied in the context of this strategy for escaping immune detection. A conformal coating of hydrogen-bonded poly(N-vinylpyrrolidone)/tannic acid (PVPON/TA) multilayer film was applied according to the protocol in [29]. First, islets were pelleted in 15 mL Eppendorf centrifuge tubes and washed twice with islet culture media. PVPON was allowed to adsorb onto islet surfaces from a 1 mg mL^−1^ solution (RPMI 1640) for 8 min on a circular roller, followed by the deposition of TA layer from a freshly dissolved 0.3 mg mL^−1^ solution (pH = 7.4) for 8 min. After each deposited layer, islets were collected by centrifugation for 1 min at 1000 rpm and rinsed with RPMI 1640. Islets were encapsulated with 4 bilayers of (PVPON/TA) with tannic acid on the outer layer. All solutions were filter-sterilised (0.22 μm pore size) with polystyrene non-pyrogenic membrane systems (Corning (Somerville, MA, USA)).

### 2.2. Islet Transplantation

Diabetes was induced in 8–10-week-old C57 BL/6Ausb mice by intravenous injection of alloxan tetrahydrate (Sigma-Aldrich (St. Louis, MO, USA)) (20 mg/mL), 110 mg/kg body weight in injectable grade water. Only mice with blood glucose ≥20 mmol/L over two consecutive readings were eligible as transplant recipients. Islets were isolated from the pancreas of donor mice and transplanted into syngeneic recipients [30]. Islets were prepared from the pooled pancreata of three donor mice (ensuring reliably sufficient numbers) and 100 hand-counted islets were transplanted into individual recipient mice. Donors and recipients were female. Islets were either control (isolated immediately) or exposed to 60 min of warm ischaemia [31]. The kidney was accessed using left flank incision and brought into the wound by gentle blunt dissection. A small nick was made in the kidney capsule at the inferior renal pole, and islets were deposited toward the superior pole. Blood glucose levels were monitored daily for seven days, then every second day up to thirty days. Two replicates were performed.

### 2.3. Hyperspectral Fluorescence Microscopy

Hyperspectral (wide-field) fluorescence microscopy used an Olympus IX83 microscope with a NuVu electron-multiplying charge-coupling device camera (EMCCD, hnu1024). LED illumination produced excitation at 325, 339, 343, 356, 366, 373, 377, 381, 384, 388, 393, 396, 400, 403, 408, 414, 425, 431, and 438 ±5 nm, while emission was detected using filters at 414, 451, 575, and 594 ±20 nm. A full detailing of excitation/emission channels is given in Appendix A, along with spectral images from each channel. The objective was 40× oil objective lens (UAPON340, Olympus, Center Valley, PA, USA). Imaging was carried out using a warm stage at 37 °C with cells in Hank’s Balanced salt solution (HBSS; Gibco, Waltham, MA, USA), which is non-fluorescent. The hyperspectral system was calibrated using a calibration fluid, which is a mixture of NADH and flavins. The calibration fluid was carefully adjusted so that its spectrum was detected across all spectral channels used in hyperspectral image acquisition. The calibration process involved capturing hyperspectral images of the calibration fluid and measuring its excitation and emission spectra separately using a fluorimeter. The obtained spectra were then used to correlate the hyperspectral images with fluorescence spectra measured on the fluorimeter as our reference, enabling the correct identification of different fluorophores in the images.

### 2.4. Image Preparation and Analysis

Image preparation was carried out [32,33] to remove image artefacts (i.e., Poisson’s noise, dead and saturated pixels, illumination curvature, background fluorescence). Regions of interest were segmented from the channel images to produce single-region images, and a variety of colour intensity features were extracted. These features included mean channel intensity and their associated statistical measures, such as channel intensity ratio [34]. Further, features related to the histogram of the cell images, such as pixels’ standard deviation and skewness, were also considered, which characterised the colour distribution of an image [35]. 

### 2.5. Modelling and Unmixing

To assess optimal separation in each case under consideration, data points representing multidimensional feature vectors for all islets or cells were projected onto an optimal (for this case) two-dimensional (2D) space created by discriminative analysis [36]. This space maximised between-group distance while minimising within-group variance, and it was spanned by two canonical variables equal to a selected linear combinations of cellular features [37]. Finally, a classifier was employed to predict the pre-defined region labels [38]. Receiver operating characteristic (ROC) area under the curve (AUC) analysis was used to quantify the accuracy of models (values of 1.0 indicate perfect accuracy, values of 0.5 indicate no better accuracy than chance). In Section 3.6, conventional, common approach of principal component analysis (PCA) was applied to the same multidimensional feature vectors for unsupervised assessment. 

In this analysis, a linear mixing model was used to identify present fluorophores in the data through spectral unmixing. Such a model assumes that each pixel contains a linear combination of distinct endmember spectra, which are weighted corresponding to the concentration of the molecules responsible for these component spectra (referred to as abundance) [2]. In this work, RoDECA, an unsupervised unmixing algorithm, was used to identify the endmember spectra and their respective abundance across the dataset. RoDECA is suitable for unmixing highly mixed datasets, including where there are no pure pixels presents in the biological samples [20]. Specific details of how RoDECA works are found in [32]. In this analysis, we specifically identified fluorophores NAD(P)H, flavins, cytochrome-C, and collagen-I, by comparing the extracted spectra by the known spectra of these pure fluorophores. We have previously shown the accuracy of RoDECA in being able to discriminate individual fluorophores in the presence of overlapping spectra and image noise [39]. Measurements of the fluorescence spectra of pure compounds (NADH, FAD, collagen-I, and Cyt) were taken and compared to the extracted endmember spectra. As per our previous studies, we created a complete excitation–emission matrix (EEM) of NADH, FAD, cytochrome-C, and collagen-I, to cross-check extracted spectra with the reference EEM.

### 2.6. Statistical Analysis

Statistical analysis was carried out in Matlab (R2017b). As the data did not meet the assumptions of parametric testing (normal distribution, equal variance), group comparisons were made using the Mann–Whitney U test, which ranks data to compare between group differences. Data are presented as median values and 95% confidence intervals. Groups were accepted as being significantly different at an alpha value of 0.05.

## 3. Results

Islets were exposed to ROS damage, hypoxia, pro-inflammatory cytokines, and warm ischaemia. In all cases, the spectral signals for the fluorophores NAD(P)H, flavins, collagen I, and cytochrome-C were identified following unmixing using RoDECA (Appendix A). These four fluorophores have had their spectral characteristics identified and assigned in a fluorophore reference bank of purified fluorophores maintained at physiological concentrations and pH [20,32,40]. Discriminant analysis was used to investigate whether a hyperspectral signal sensitive to the presence of a viability-compromising exposure could be constructed. This was repeated for single-cell suspensions from disaggregated islets, as this strategy would have greater clinical utility. Unmixing was also undertaken for single cells where findings paralleled the results for whole islets (Appendix A). 

### 3.1. ROS Damage

ROS damage by menadione created a visible change in the spectral morphology of many islets (Figure 1). This effect was not apparent in all islets; however, there was no overlap in the clusters formed from canonical variables our model developed from the hyperspectral data. As such, we were able to achieve full discrimination of the two groups (Figure 2A,B), as indicated by an AUC of 1.00. Similar accuracy was achieved for single cells (AUC = 0.99, Appendix A) despite the loss of the whole islets’ distinctive morphology. Unmixing in whole islets was able to identify spectral patterns corresponding to the autofluorescent coenzymes NAD(P)H and flavins, the structural protein collagen-I, and the mitochondrial protein cytochrome-C. Additionally, the redox ratio was calculated as the ratio of unmixed abundances of NAD(P)H and flavin signals. The only statistically significant difference was for redox ratio, which was elevated in islets exposed to ROS damage (Figure 3).

### 3.2. Hypoxia

The induction of hypoxia in islets through DMOG exposure did not result in an obvious disruption in islet morphology. The clusters, additionally, could not be separated to the same degree (Figure 2C), resulting in lower accuracy for discrimination (Figure 2D, AUC = 0.69). A minor difference was seen for single cells, which could be separated with AUC = 0.72 (Appendix A). Unmixing showed a significant elevation in redox ratio (Figure 4D), but here the increase for NAD(P)H signal was also significant (Figure 4A). 

### 3.3. Inflammatory Cytokines

The inflammation model was more complex, with three comparisons. Islets exposed to moderate pro-inflammatory signalling (4 h) were discernible from pristine islets with AUC = 0.79 (Figure 5A,B), while discrimination for islets exposed to major pro-inflammatory signalling (24 h) were discriminable with an AUC of 0.94 (Figure 5C,D). The apparent presence of a dose response supports that the hyperspectral imaging of autofluorescence was detecting the islets’ response to exposure to the proinflammatory cytokines. The two groups of exposed islets were discriminable from one another with AUC = 0.95 (Figure 5E,F). Single cells from islets exposed to major and moderate pro-inflammatory signalling could be discriminated from cells from pristine islets with AUC 0.94 and 0.98 (Appendix A), respectively. Unmixing analyses show significant increases in NAD(P)H signal for both 4 h and 24 h of exposure (Figure 6A). Flavin signal was not significantly affected (Figure 6B); however, redox ratio (RR, NAD(P)H/Flavin) was increased for 24 h cytokine exposure compared to 24 h control (Figure 6D). For both exposure times, lower collagen-I signal was detected in cytokine-treated islets (Figure 6C); however, cytochrome-C signal was not affected. Comparisons were also made between 4 h control and 24 h, control as well as 4 h cytokine and 24 h cytokine exposure. The only effect was for NAD(P)H signal, where the difference between the two periods was *p* = 0.05. 

### 3.4. Warm Ischaemia

Islets were exposed to warm ischaemia to directly model the damage which human islets used for transplantation may experience prior to pancreatic retrieval (Figure 7). Islets with moderate exposure to warm ischaemia could be discriminated with accuracy of AUC = 0.82 (Figure 7A,B, AUC = 0.82). This was improved, however, in islets with major exposure (AUC = 0.92, Figure 7C,D). The discrimination of islets with moderate compared to major exposure was AUC = 0.80 (Figure 7E,F). Three-way discrimination, which sorted the islets into each of the three groups (control, moderate, and major), achieved AUC = 0.76 (Appendix A). This suggests hyperspectral imaging of autofluorescence could give a scaled indication of islet preparation viability. Results for single cells had comparable AUC values, although the accuracy was similar for moderate and extreme exposure, with moderate exposure being very slightly ahead (AUC = 0.86 and 0.84, Appendix A).

Unmixing showed a significant increase in NAD(P)H signal for 60 min of warm ischaemia compared to both control and 30 min (Figure 8A). The signal for flavins was significantly reduced by both 30 and 60 min warm ischaemia compared to the controls, but not relative to each other (Figure 8B). This was reflected by warm ischaemia significantly increasing RR after both 30 and 60 min (Figure 7D). Effects for the collagen-I signal were highly significant (Figure 8C), but without a clear pattern, with 30 min increasing collagen-I relative to control, and 60 min being significantly lower than control. In the assessment of single cells, wherein the extracellular matrix had been further disrupted through exposure to EDTA for disaggregation, no significant differences were found between groups (Appendix A). The cytochrome-C signal was significantly elevated by 60 min of warm ischaemia compared to both controls and 30 min (Figure 8E).

### 3.5. Encapsulation

The encapsulation of islets in a multilayer conformal film was investigated to determine whether the hyperspectral imaging of islet autofluorescence could still detect viability-compromising insults in this context. First, we investigated whether the presence of the conformal coating had an impact on the hyperspectral properties of the islets by comparing the profiles of encapsulated islets to those which had been taken through the encapsulation process without the application of the conformal coating (unencapsulated islets). They were able to be discriminated with an AUC = 1.00. This perfect accuracy was not surprising given the visible impact of the conformal coating on the spectral properties of the islets (Appendix A). Unencapsulated islets were also discriminated from pristine control islets at AUC = 1.0 (Appendix A), suggesting that the encapsulation process itself had a major effect on islet characteristics. This raised the question as to whether other sources of potentially compromising insult could still be detected in encapsulated islets. To assess this, we reperformed the previously described cytokine experiment. If encapsulation material had too large of an impact on the hyperspectral characteristics of islets, it would overshadow our ability to use hyperspectral microscopy to detect damage. This was not the case, however, as 4 h and 24 h of exposure to pro-inflammatory cytokines could be discriminated with AUC values of 0.73 and 0.89, respectively, which are comparable to previous values (Appendix A). The other implications of these findings are that the impact of the encapsulation process is not so major that the comparatively minor impact of 4 h exposure to cytokines cannot be detected, and that, not surprisingly, the encapsulation envelop does not exclude our proinflammatory cytokines.

### 3.6. Prediction of Transplant Success

Glucose control after transplantation was assessed over 30 days (Figure 9A). All transplanted control islets maintained consistent glucose control from day 1 to the end of follow up. Three of the five islet preparations subjected to warm ischaemia (Figure 9A) did not restore glucose control at day 30, while two did. In both cases, the ischaemic islets which restored glucose control had partial function in the initial period (<5 days), but maintained normo-glycemia by the completion of monitoring (Figure 9A). The inclusion of islets exposed to warm ischaemia in both groups (functional/non-functional) is important, as it means that this analysis is not synonymous with the discrimination of islets exposed to warm ischaemia and those not.

Unsupervised PCA was applied to multidimensional feature vectors for all islets to investigate their emergent sorting (Figure 9). Groups could be seen with a bias for clustering islets from functional and non-functional preparations (Figure 9B) more so than control and ischaemic preparations (Figure 9B). Principal component 2 emerged from this assessment (Figure 9C) as a potential informative “viability score”. Functional and non-functional preparations could be separated in all cases by their being above or below a threshold value for this “viability score” (Figure 9C). An ROC curve of individual islets had lower accuracy (AUC = 0.81, Figure 9D); however, these preparations should be expected to be heterogenous, with overall functional preparations containing a minority of compromised islets and non-functional preparations containing some healthy islets. These findings are particularly meaningful, as they are the result of unsupervised analysis—that is, the model was not structured to separate preparations that restored glucose control from those which did not; instead it was “blind” to group membership and the distinction emerged when it sorted like with like. Data for single cells, which are more vulnerable to motion artifacts, had too high a variance for unsupervised assessment to be undertaken.

## 4. Discussion

There are several assays for post-transplant islet function; however, they have all been found to be insufficiently reliable predictors for robustly informing clinical decision making [4,5,6,7,8,9]. Optimally, technologies that interrogate an islet preparation’s functional capacity will be non-invasive [4] in order to maximally preserve islet numbers in clinical contexts and enable direct follow-up experiments in research contexts. The hyperspectral microscopy of autofluorescence enabled the differentiation of individual islets that were damaged, as compared to those in a pristine resting state. Additionally, it was able to discern the individual components of damaging stimuli, and so could discern islets exposed to ROS damage, pro-inflammatory cytokine signalling, or warm ischaemia. Strong accuracy (AUC > 0.9) was achieved for detecting islets exposed to ROS, pro-inflammatory cytokine signalling, or warm ischaemia. Further refinements may improve the detection of hypoxia damage. Further, an unsupervised algorithm prospectively identified which islet preparations restored glucose control in diabetic mice. This nominally outperforms any prior technology [4,5,6,7,8,9]; however, further replication is needed to demonstrate reliability. These data show hyperspectral microscopy has strong potential to be translated to assess islet viability and inform clinical decision making with novel information on the likelihood of transplant success. As well as reducing patient burden, the withdrawal of immunosuppression following failed islet transplants results in heightened sensitisation to human leukocyte antigens [41,42]. This increases the importance of avoiding islet preparations with a low likelihood of successful transplantation.

### 4.1. Islet Disaggregation

The 3D nature of islets could interfere with their emission spectrum via absorption. Single cell imaging avoids this possibility, but at the expense of structural information. However, when islets were disaggregated, the accuracy of the hyperspectral models was not compromised. Due to their size, only one to three islets can be imaged at a time, which would reduce the efficiency of assessment. To investigate a strategy to overcome this limitation, we homogenised suspensions of single cells to enable the collection of 50–80 datapoints per field-of-view—presumably representative of all disaggregated islets. This allowed the rapid collection of a characteristic dataset with a lower image preparation burden, decreasing the time taken for assessment. These findings also demonstrate the potential of this technology to be applied for emerging beta cell replacement strategies, such as those based on stem cell culture and differentiation, beyond deceased donor islet transplantation. 

### 4.2. Biomarkers

The redox ratio was significantly increased in islets with compromising exposures across all conditions, with the exception of moderate exposure to pro-inflammatory cytokines, where statistical significance was not reached (*p* = 0.11). An associated significant increase in NAD(P)H was also frequently observed, sometimes accompanied by a reduction in flavins. These effects are well supported by the literature. ROS generation from menadione exposure in pancreatic β-cells is driven by elevated NADH [43], and NADPH maintains systems which defend against cellular ROS damage [43]. Hypoxia necessitates greater reliance on anerobic glycolysis, which is marked by a shift from flavins towards NAD(P)H [44]. The activation of immune cells by inflammatory signalling has been linked to glycolysis and increased redox ratio [45]; however, our observation of the redox ratio in pancreatic islets being increased by pro-inflammatory signalling appears to be novel. Increasing levels of NADH in blood have also been observed with increasing post-mortem interval [46]. Furthermore, both regional and global myocardial ischaemia induced in isolated rat hearts resulted in a rapid, substantial increases in the intensity of NADH autofluorescence [47]. 

The potential of elevated redox ratio to act as a consistent biomarker of islet viability is further supported by the observation of increased NAD(P)H relative to flavins in cells undergoing apoptosis [48]. Furthermore, cell metabolism shifting away from oxidative phosphorylation and towards anerobic glycolysis, indicated by the elevated redox ratio [44], represents a critical junction in islet physiology. This is because the decreased production of ATP from glycolysis relative to oxidative phosphorylation cripples glucose-triggered insulin secretion, rendering islets non-responsive to fluctuating glucose in the external milieu [11,49].

We also isolated the spectral signal of cytochrome-C, although the only significant findings were made for warm ischaemia where it was elevated in islets with major exposure relative to pristine islets with moderate exposure. Cytochrome-C is generally located between the inner and outer mitochondrial layers, where it is an essential component of the electron transport chain. The link between islet viability and Cytochrome-C may be via its connection to apoptosis. Cytochrome-C being released from the mitochondria into the cytosol is a primary driver of apoptosis, as it activates a caspase cascade that commits the cell to the death process. This change in localisation is generally not expected to alter the relative strength of its total spectral signal; however, the upregulation of cytochrome-C production has also been observed to accompany apoptosis [50].

### 4.3. Translation

Islets collected for transplantation are exposed to a number of viability-compromising insults. We have shown that hyperspectral microscopy has the sensitivity to discern these insults by the spectral signature they impart on an islet. Furthermore, we demonstrated that the hyperspectral imaging of islets is able to produce a “viability score” which reflects the ability of otherwise-compromised islets to yield good glucose control after transplantation into diabetic mice. Such a score is critically required for clinical islet transplantation where islets come from deceased donors and are known to have been subjected to multiple insults, including warm ischaemia, hypoxia, oxidative stress, and cytokine injury. 

Potential limitations of this technology primarily relate to translation from mouse to humans, including both biological variation between species and the heterogeneity introduced by the clinical environment (e.g., variable insulin secretion function, patient and donor specific factors). The increased presence of acinar tissue seen in human preparations compared to mouse preparations should not present a challenge, as this tissue is morphologically distinct from islets under normal brightfield microscopy [51]. Some factors could be simplified, however, an example being that islets for human transplant are only ever prepared from a single source, whereas our experimental design required pooling. As such, donor specific factors such as sex (exposure to estradiol has been shown to protect beta cells from apoptosis [52]) could be factored into predictive modelling. It should be noted that the viability signatures established here in mice cannot be directly extrapolated to humans due to differences in physiology and metabolomics. Future research, beyond extension to humans, could involve clarifying the system’s sensitivity to combinations of insults, as well as the lower bound of its sensitivity, in order to understand its clinical translatability. Moreover, the reliability of the “viability score” should be reinforced by further replications, including for a greater number of insults. The hyperspectral images took approximately six minutes to collect, but this would be significantly reduced in translation through the use of a task-specific instrument and protocol which could have uninformative channels removed and a wider field of view. The absence of phototoxicity from the excitation illumination wavelengths affecting islet function and viability should also be validated in future research. However, as we have shown that this technology does not compromise in vitro cultured embryos—a notoriously fragile system—such an effect is unlikely [53]. 

## 5. Conclusions

This study offers a clear pathway for grading islet preparations and predicting their future performance. The strength of our findings is reinforced by the use of fully unsupervised assessment. An additional strength is that this approach could be applied to alternate sources of beta cells, including the quality assessment of xenogenic islets [54], and potentially stem-cell-derived islets, where distinguishing differentiated functional beta cells from undifferentiated cells would be advantageous [55]. For application to adult cadaveric islets, as currently used in clinical practice, the successful translation of this technology has the potential to give a non-invasive indication of islet viability, prior to transplantation, which would inform clinical decision making and enable patients to be spared transplantation attempts with no potential to reduce their dependence on exogenous insulin.

## Figures and Tables

**Figure 1 cells-12-02302-f001:**
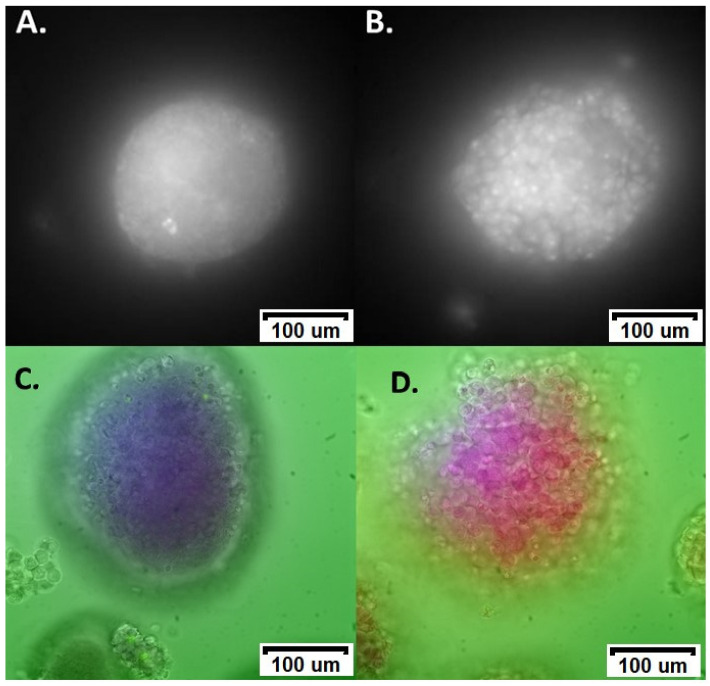
Representative spectral images of islets subjected to ROS. (**A**) Control islet autofluorescence; (**B**) ROS damage islet autofluorescence. (**C**) False colour principal component analysis (PCA) image superimposed with brightfield image of control islets. (**D**) False colour PCA image superimposed with brightfield image of ROS damaged islets. The false colours in the PCA image visualise compressed pixels using different channels resulting from the PCA analysis, to enhance visual understanding of data clusters obtained from multivariate analysis.

**Figure 2 cells-12-02302-f002:**
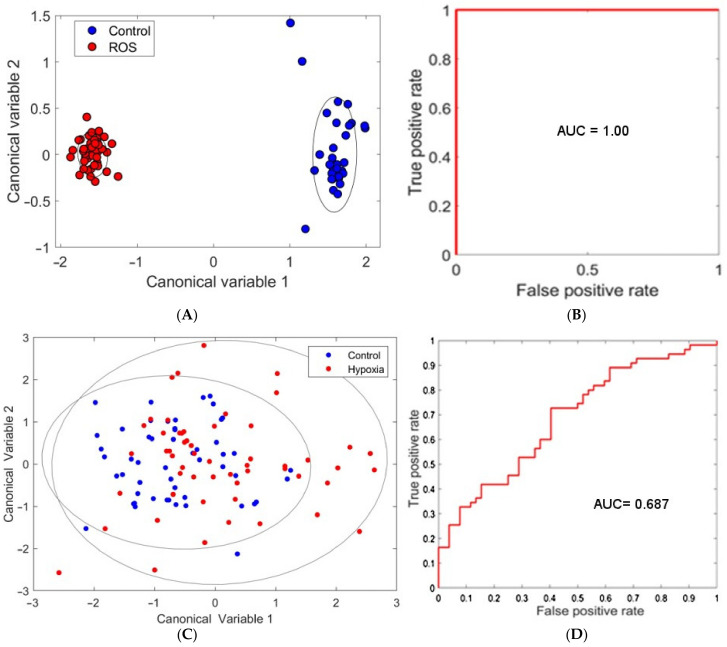
ROS damage induced by exposing islets to menadione. (**A**) Canonical discriminant analysis for the discrimination of control (blue), and exposed (red). (**B**) ROC curve showing complete discrimination (AUC = 1.0). Corresponding 95% confidence intervals are drawn as ellipses. Hypoxia damage induced by exposing islets to DMOG. (**C**) Canonical discriminant analysis for control (blue) islets and hypoxic (red) islets. Corresponding 95% confidence intervals are drawn as ellipses. (**D**) ROC curve showing partial discrimination (AUC = 0.69).

**Figure 3 cells-12-02302-f003:**
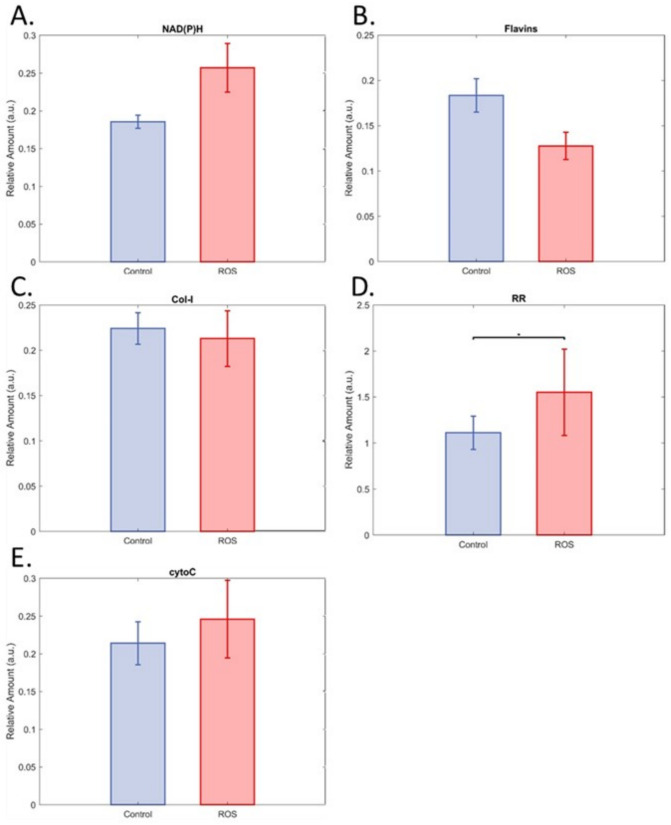
Impact of ROS exposure (induced by menadione) on relative fluorophore levels. (**A**) NAD(P)H (indicating the combined signals of NADH and NADPH, of which NADH is most plentiful), (**B**) flavins (indicating the combined signals of flavin family members, of which the metabolic coenzyme to NADH, FAD, is the most plentiful), (**C**) Col-I (collagen-I), (**D**) redox ratio (RR = NAD(P)H/flavins), and (**E**) cytochrome-C. Data are means with 95% CI, * shows *p* = 0.0444. In all cases, n = 24 for control and 25 for ROS.

**Figure 4 cells-12-02302-f004:**
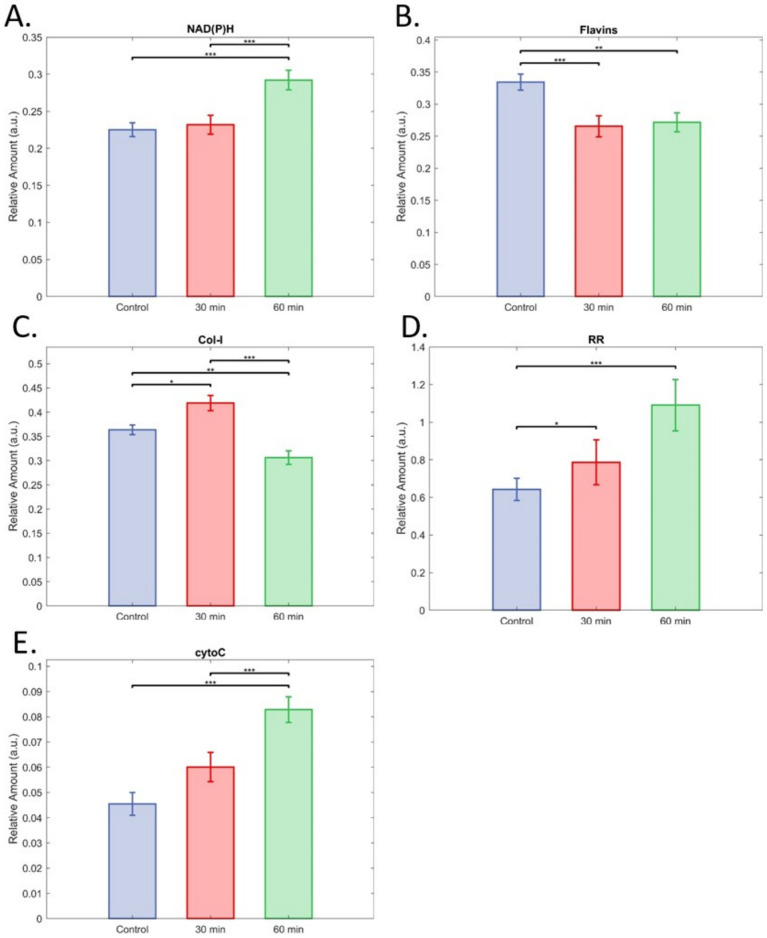
Impact of hypoxia-relative fluorophore levels. (**A**) NAD(P)H, (**B**) flavins, (**C**) Col-I (collagen-I, (**D**) redox ratio (NAD(P)H/flavins), and (**E**) cytochrome-C. Data are means with 95% CI, * shows *p* < 0.05, ** shows *p* < 0.01, *** shows *p* < 0.001. In all cases, n = 36 for control and 32 for hypoxia.

**Figure 5 cells-12-02302-f005:**
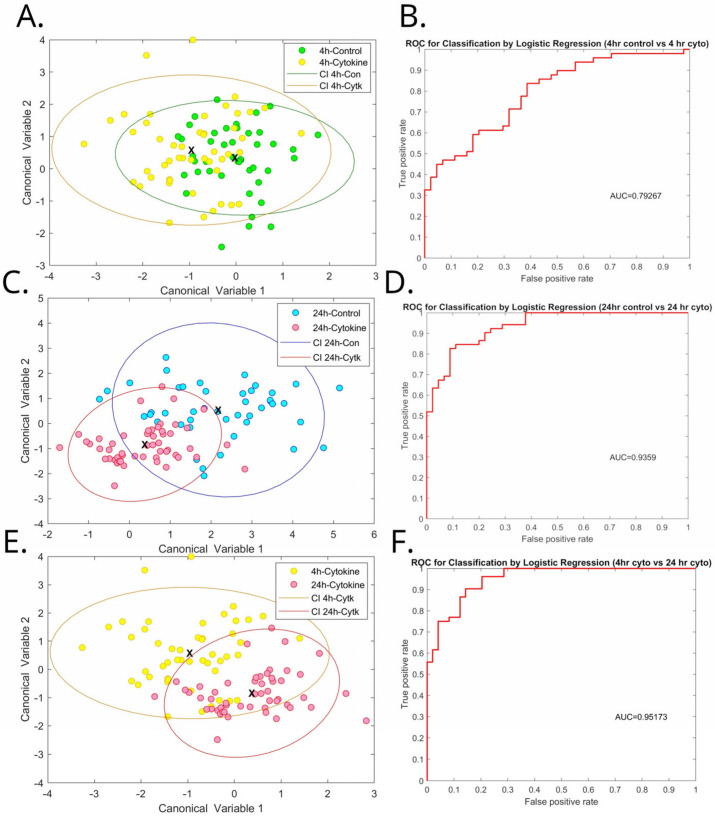
Hyperspectral signature of moderate (4 h) and major (24 h) pro-inflammatory signalling. Canonical discriminant analysis for differentiating islets exposed moderate pro-inflammatory signalling from pristine islets with the corresponding ROC curve is shown in (**A**,**B**), the same for differentiating major pro-inflammatory signalling from pristine is shown in (**C**,**D**), and the differentiation of moderate from major is shown in (**E**,**F**). Corresponding 95% confidence intervals are drawn as ellipses.

**Figure 6 cells-12-02302-f006:**
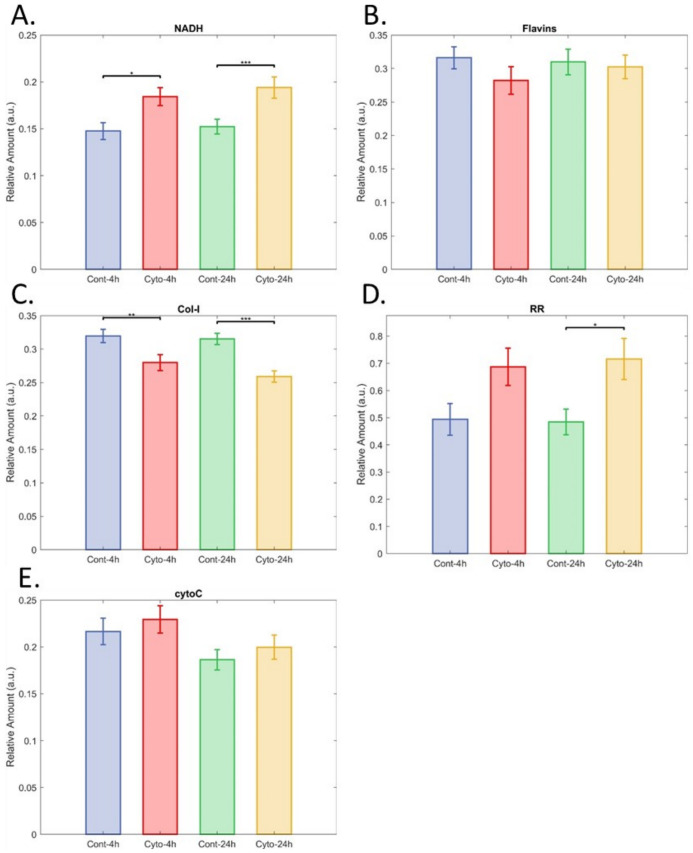
Impact of pro-inflammatory cytokine exposure on relative fluorophore levels. (**A**) NAD(P)H, (**B**) flavins, (**C**) Col-I (collagen-I), (**D**) redox ratio (NAD(P)H/flavins), and (**E**) cytochrome-C. Data are means with 95% confidence interval, in all cases n = 34 for control 4 h (Cont-4 h), 48 for cytokine exposure 4 h (Cyto-4 h), 36 for control 24 h (Cont-24 h), and 50 for cytokine 24 h (Cyto 24 h). *, **, and *** indicate significant differences at *p* < 0.05, 0.005, and 0.0001.

**Figure 7 cells-12-02302-f007:**
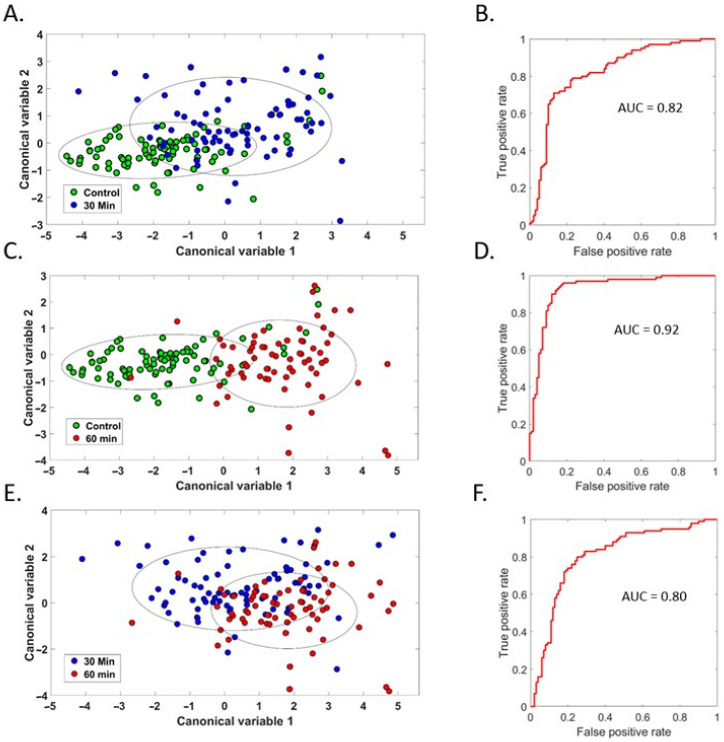
Hyperspectral signature of moderate (30 min) and major (60 min) exposure to warm ischaemia. Canonical discriminant analysis for differentiating islets exposed to moderate warm ischaemia from pristine islets with the corresponding ROC curve is shown in (**A**,**B**), the same for differentiating major warm ischaemia from pristine islets is shown in (**C**,**D**), and the differentiation of moderate from major is shown in (**E**,**F**). Corresponding 95% confidence intervals are drawn as ellipses.

**Figure 8 cells-12-02302-f008:**
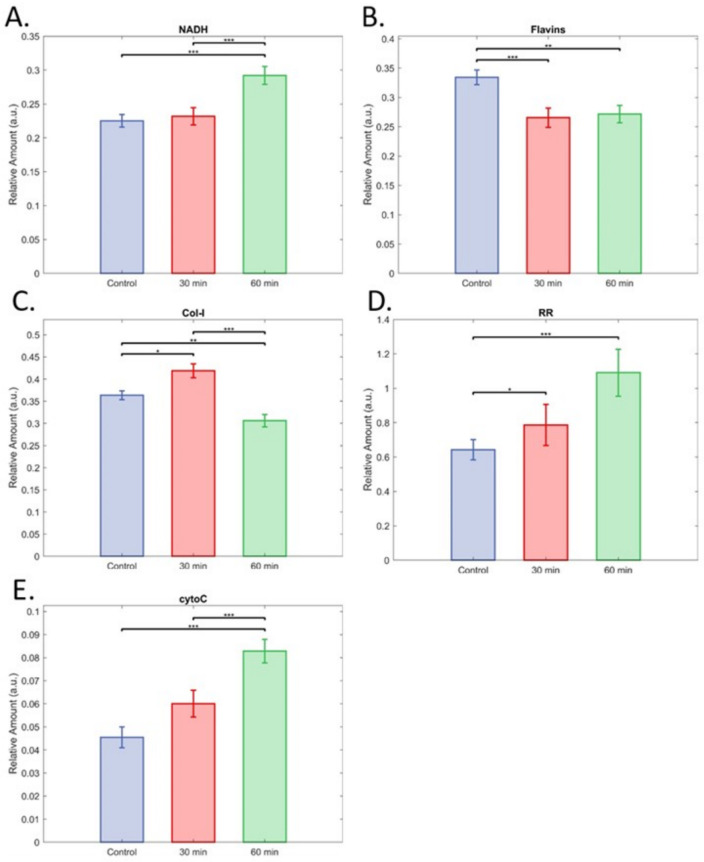
Impact of warm ischaemia on relative fluorophore levels. (**A**) NAD(P)H, (**B**) flavins, (**C**) Col-I (collagen-I), (**D**) redox ratio (RR), and (**E**) cytochrome-C. Data are means with 95% CI. In all cases n = 79 for control islets, 72 for islets from pancreata left in mice for 30 min post mortem, and 62 for islets left in mice for 60 min post mortem. *, **, and *** indicate significant differences at *p* < 0.05, 0.005, and 0.0001.

**Figure 9 cells-12-02302-f009:**
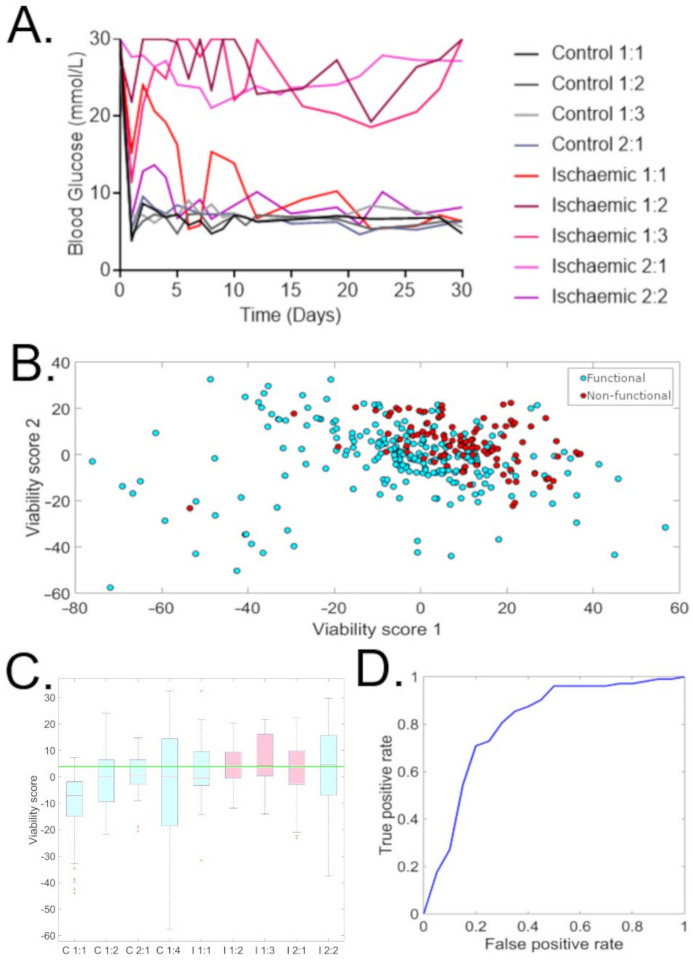
Unsupervised assessment of hyperspectral data for the transplanted islets. Principal component analysis (PCA) was applied for multidimensional feature vectors for all islets. (**A**) Blood glucose control after islet transplantation into mice. Islet sets labelled as treatment group (control or Ischaemic), replicate number: number within replicate. (**B**) Islets which restored glucose control (functional, blue dots) plotted against islets which did not restore glucose control (red dots) in the space spanned by the first and second PCA component. (**C**) Box plots of the values of the second PCA component for the different islet sets where C = control and I = ischaemic treated group. A threshold line was able to be drawn, which lay above the median value for all functional islet preparations and below the median value of all non-functional islet preparations, giving 100% accuracy for discrimination at the group level. (**D**) ROC curve for individual islets, AUC = 0.81.

## Data Availability

Data will be shared with interested third parties on reasonable request.

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
