# Peer review of "Pancreatic Islet Viability Assessment Using Hyperspectral Imaging of Autofluorescence"

_cells, 2023, doi:10.3390/cells12182302_

Round 1

Reviewer 1 Report

Strengths:

  1. The manuscript offers a complete introduction and skillfully outlines the existing restrictions on tests for islet function following transplantation. The later introduction of hyperspectral microscopy is essentially made possible by the establishment of this fundamental framework.
  2. A thorough explanation of hyperspectral microscopy's capabilities, including its potential to discern between injured and unaltered islets, is provided in the technique section.
  3. The conclusions made are backed up by solid data. The focus on the high redox ratio as a potential biomarker is particularly significant, and both the study's findings and the literature it references provide strong support for this assertion.
  4. The authors' recognition of areas requiring additional research gives the manuscript depth and balance and demonstrates a noteworthy level of academic rigor.

Areas for Improvement:

  1. The manuscript's structure needs to be improved, especially in the discussion section. The presence of intricate sentences could potentially impede comprehension, and the incorporation of subheadings might enhance the section's structure and readability.
  2. While the manuscript briefly addresses the challenges of extrapolating findings from mouse models to human contexts, a more comprehensive exploration of this topic would enrich the discussion, given its pivotal clinical implications.

Recommendations:

  1. I recommend a restructuring of the discussion section to foster a more coherent narrative flow.
  2. Furthermore, I encourage the authors to provide a more in-depth analysis of the intricacies and potential challenges inherent in translating findings from mouse models to human clinical scenarios.

none

Author Response

Reviewer 1

  1. The manuscript offers a complete introduction and skillfully outlines the existing restrictions on tests for islet function following transplantation. The later introduction of hyperspectral microscopy is essentially made possible by the establishment of this fundamental framework.
  2. A thorough explanation of hyperspectral microscopy's capabilities, including its potential to discern between injured and unaltered islets, is provided in the technique section.
  3. The conclusions made are backed up by solid data. The focus on the high redox ratio as a potential biomarker is particularly significant, and both the study's findings and the literature it references provide strong support for this assertion.
  4. The authors' recognition of areas requiring additional research gives the manuscript depth and balance and demonstrates a noteworthy level of academic rigor.

Areas for Improvement:

  1. The manuscript's structure needs to be improved, especially in the discussion section. The presence of intricate sentences could potentially impede comprehension, and the incorporation of subheadings might enhance the section's structure and readability.
  2. While the manuscript briefly addresses the challenges of extrapolating findings from mouse models to human contexts, a more comprehensive exploration of this topic would enrich the discussion, given its pivotal clinical implications.

Recommendations:

  1. I recommend a restructuring of the discussion section to foster a more coherent narrative flow.

Per your above recommendation we have added a series of subheadings to aid the interpretability of the Discussion section. We have additionally revised the language to improve interpretability.

  1. Furthermore, I encourage the authors to provide a more in-depth analysis of the intricacies and potential challenges inherent in translating findings from mouse models to human clinical scenarios.

The following point has been added to the consideration of the translation of this work from mouse to human:

“It should be noted that the viability signatures established here in mice cannot be directly extrapolated to humans due to differences in physiology and metabolomics.”

Moreover an additional point regarding the comparative importance of donor specific factors has been made in response to Reviewer 3.

Reviewer 2 Report

In this well-written manuscript, Campbell et al. evaluate an innovative, hyperspectral in vitro assessment of isolated murine islets of Langerhans based on intrinsic islet autofluorescence characteristics/signatures (of NADH, flavins, collagen and cytochrome-c). The authors compare the hyperspectral valuation/signals of ‘pristine’ islets to that of islets exposed to oxidative stress, cytokine exposure and warm ischemia; insults which have limited the success of clinical islet transplantation. Furthermore, in a blinded fashion, the authors evaluated the autofluorescence signatures for their predictive capacity of subsequent post-transplant function, in mice. Overall, the manuscript is concise and well written, and the data presented does support the author's conclusions that a non-invasive hyperspectral assessment of islet preparations may be a promising tool to predict in vivo function. Whether these observations translate to clinical relevance, has yet to be demonstrated but may prove to be a useful tool to predict the in vivo function of primary and stem cell derived-human islets.

Author Response

Reviewer 2

In this well-written manuscript, Campbell et al. evaluate an innovative, hyperspectral in vitro assessment of isolated murine islets of Langerhans based on intrinsic islet autofluorescence characteristics/signatures (of NADH, flavins, collagen and cytochrome-c). The authors compare the hyperspectral valuation/signals of ‘pristine’ islets to that of islets exposed to oxidative stress, cytokine exposure and warm ischemia; insults which have limited the success of clinical islet transplantation. Furthermore, in a blinded fashion, the authors evaluated the autofluorescence signatures for their predictive capacity of subsequent post-transplant function, in mice. Overall, the manuscript is concise and well written, and the data presented does support the author's conclusions that a non-invasive hyperspectral assessment of islet preparations may be a promising tool to predict in vivo function. Whether these observations translate to clinical relevance, has yet to be demonstrated but may prove to be a useful tool to predict the in vivo function of primary and stem cell derived-human islets.

We thank the reviewer for their encouraging commentary. We have clarified in our conclusion that impacts on clinical care would require further translational research:

“For application to adult cadaveric islets, as currently used in clinical practice, the successful translation of this technology has the potential to give a non-invasive indication of islet viability, prior to transplantation, which would inform clinical decision making and enable patients to be spared transplantation attempts with no potential to reduce their dependence on exogenous insulin.”

Reviewer 3 Report

The manuscript by Campbell et al. establishes a non-invasive system to assess the islet viability for transplantation. They accomplished this by using the hyperspectral imaging detection of autofluorescence from the islets exposed to ROS damage, hypoxia, inflammation, or ischemia. The assay accurately discriminated the compromised islets from the controls in all conditions except for hypoxia. The authors observed a consistent increase in the redox ratio across all insult conditions. Furthermore, their unsupervised analysis of the islet transplantation experiment introduced “viability score”, and upon further validation, this work could potentially provide a pre-transplantation measurement of islet viability and predict transplantation success. This work provided new data to the field and thus is suitable for publication in Cells with revisions.

Major comment

Line 90: The islets in the study were isolated from both male and female C57BL/6Ausb mice and mixed for use. It is known that estradiol protects beta cells from apoptosis (Mauvais-Jarvis et al., Adv Exp Med Biol, 2017). Failure to consider the sex differences and the choice of mice used for islet isolation could lead to increased variability and decreased reproducibility. Authors should consider restricting to either female or male mice. This is also more relevant to the clinical setting with only one islet donor pancreas.

Line 100: It appeared that the authors maintained the control islets only in the culture media, without giving them control treatment. For example, menadione is soluble in either chloroform or DMSO. It requires the control treatment to the same solvent concentration.

In the clinical setting, as agreed by the authors, the isolated human islets could potentially encounter a variety of insults that compromise the islet viability. The authors designed the experiment and recreated these insults individually to examine the ability of hyperspectral imaging to assess the islet viability. Additional conditions that combine multiple insults, especially with treatment at a much lower concentration/shorter timeframe, could be added to examine the assay’s reliability.

To confidently establish the “viability score”, more replications and conditions (more than just ischemia) for islet transplantation are needed.

Minor comments

Line 310: Please indicate where the data is for the three-way discrimination.

Line 363, 365, 368: Fig.9A, not Fig.9E, were described here.

The color codes for Fig.9A made it very difficult to distinguish respective ischemia group members.  

Fig.9 panel labeling was wrong.

Author Response

Reviewer 3

The manuscript by Campbell et al. establishes a non-invasive system to assess the islet viability for transplantation. They accomplished this by using the hyperspectral imaging detection of autofluorescence from the islets exposed to ROS damage, hypoxia, inflammation, or ischemia. The assay accurately discriminated the compromised islets from the controls in all conditions except for hypoxia. The authors observed a consistent increase in the redox ratio across all insult conditions. Furthermore, their unsupervised analysis of the islet transplantation experiment introduced “viability score”, and upon further validation, this work could potentially provide a pre-transplantation measurement of islet viability and predict transplantation success. This work provided new data to the field and thus is suitable for publication in Cells with revisions.

Major comment

Line 90: The islets in the study were isolated from both male and female C57BL/6Ausb mice and mixed for use. It is known that estradiol protects beta cells from apoptosis (Mauvais-Jarvis et al., Adv Exp Med Biol, 2017). Failure to consider the sex differences and the choice of mice used for islet isolation could lead to increased variability and decreased reproducibility. Authors should consider restricting to either female or male mice. This is also more relevant to the clinical setting with only one islet donor pancreas.

This issue has been addressed in the discussion as follows:

“Some factors could be simplified, however; an example being that islets for human trans-plant are only ever prepared from a single source whereas our experimental design re-quired pooling. As such, donor specific factors such as sex (exposure to estradiol has been shown to protect beta cells from apoptosis [52]) could be factored into predictive model-ling.”

Line 100: It appeared that the authors maintained the control islets only in the culture media, without giving them control treatment. For example, menadione is soluble in either chloroform or DMSO. It requires the control treatment to the same solvent concentration.

Control solvent concentrations were included in all control treatments. We regret that we were unclear on this point in our description of our Methods and have updated them to be clear as follows:

“Control islets were maintained in culture media with equivalent solvent used to emulsify interventions as applicable for identical time-courses.”

In the clinical setting, as agreed by the authors, the isolated human islets could potentially encounter a variety of insults that compromise the islet viability. The authors designed the experiment and recreated these insults individually to examine the ability of hyperspectral imaging to assess the islet viability. Additional conditions that combine multiple insults, especially with treatment at a much lower concentration/shorter timeframe, could be added to examine the assay’s reliability.

To confidently establish the “viability score”, more replications and conditions (more than just ischemia) for islet transplantation are needed.

We agree with the reviewer, although we would like to note that the ‘warm’ ischemia intervention was an attempt to model more multifactorial insults that come closer to the clinical experience as it does, in effect, comprise exposure to ROS, cytokines, and hypoxia. We have discussed these additional areas for experimental investigation as follows:

“Future research, beyond extension to humans could involve the clarifying the systems sensitivity to combinations of insults, as well as the lower bound of its sensitivity in order to understand its clinical translatability. Moreover, the reliability of the “viability score” would be reinforced by further replications, including for a greater number of insults.”

Minor comments

Line 310: Please indicate where the data is for the three-way discrimination.

Data for the three way discrimination is presented in Supplementary figure 1D. A note has now been included in the text of the results directing readers to it.

Line 363, 365, 368: Fig.9A, not Fig.9E, were described here.

Thank you. I have reviewed the section of text pertaining to Fig. 9 and corrected any mistakes in labelling.

The color codes for Fig.9A made it very difficult to distinguish respective ischemia group members.  

Colouring has been chosen to best enable Control groups to be distinguished from Ischemic. While acknowledging your point, we do believe that the present scheme is optimal.

Fig.9 panel labeling was wrong.

We apologise for the oversight. The labelling has now been corrected.

Round 2

Reviewer 3 Report

The authors addressed my feedback from the first round of peer review and presented additional discussion regarding the limitations of the study in its current form. As a result, I believe this manuscript can be considered for publication in Cells.